# A Case–Control Study on the Usefulness of Serum Lecithin: Cholesterol Acyltransferase Activity as a Predictor of Retained Placenta in Close-Up Dairy Cows

**DOI:** 10.3390/ani14243640

**Published:** 2024-12-17

**Authors:** Hiroki Satoh, Kyoko Chisato, Rika Fukumori, Mohamed Tharwat, Shin Oikawa

**Affiliations:** 1Veterinary Herd Health, Department of Veterinary Medicine, School of Veterinary Medicine, Rakuno Gakuen University, Ebetsu 069-8501, Hokkaido, Japan; s21161058@g.rakuno.ac.jp (H.S.); k-chisato@rakuno.ac.jp (K.C.); fukumori@rakuno.ac.jp (R.F.); 2Department of Clinical Sciences, College of Veterinary Medicine, Qassim University, P.O. Box 6622, Buraidah 51452, Saudi Arabia; atieh@qu.edu.sa

**Keywords:** cow, retained placenta, LCAT, NEFA, predictor

## Abstract

Retained placenta (RP) is one of the most periparturient diseases of dairy cows, in which the placenta is not expelled from the body within 24 h of calving. This condition increases the risk of other periparturient diseases and decreased milk production and reproductive efficiency. In order to reliably predict PR, this study focused on the activity of lecithin:cholesterol acyltransferase (LCAT) and compared its usefulness in the prepartum prediction of PR with the concentration of non-esterified fatty acids (NEFA). LCAT is an enzyme synthesized in the liver that converts free cholesterol to cholesteryl esters, which is known to be acutely related to and reduced by periparturient diseases. NEFA is a fatty acid that increases in the blood when animals are in negative energy balance and has been reported as a useful prepartum predictor of some perinatal diseases. This study was carried out at a single farm with approximately 200 Holstein parous cows from February 2010 to February 2016. Twenty-seven parous cows between 2 and 21 days (close-up stage) before their expected calving dates that developed RP (RP group) were assessed. They were compared with 60 clinically healthy cows (controls) that did not develop RP and were matched with the RP group for sampling period and parity. The results of this study were as follows: LCAT showed adequate discriminative ability of PR occurrence comparable to that of NEFA. However, LCAT or NEFA plus RFS showed higher discrimination ability than both alone. These results indicate that LCAT has a useful ability to predict the occurrence of RP.

## 1. Introduction

Retained placenta (RP) is a periparturient disease of dairy cows defined as a condition in which the placenta is not expelled from the body within 24 h of calving [1,2]. This condition increases the risk of other periparturient diseases such as uterine infection and endometritis, as well as reducing milk production and reproductive efficiency [1,3], greatly lessening the overall productivity of dairy farms [1,2]. Therefore, it is plausible that early prediction of RP could help alleviate economic losses.

A recent report stated that the direct causes of RP are uterine atony, dystocia, uterine torsion, edema of the chorionic villi from procedures such as cesarean section, endometrial epithelial necrosis and dysfunction due to uterine infection, steroid hormone imbalance, a reduced immune response, and aggravated oxidative stress in the body [4]. In addition, negative energy balance (NEB) during the dry period is suggested to be an underlying factor for the onset of RP [5]. Negative energy balance increases mobilization of non-esterified fatty acids (NEFA) from adipose tissue to the liver [6,7]. In the liver, mobilized fatty acids are either β-oxidized for ATP production or re-esterified to triglycerides (TG) to be released into the blood as very low-density lipoproteins (VLDL). During the dry period, the latter pathway tends to be favored because the hepatocytes contain sufficient glucose [8]. The bovine liver has a low capacity for secretion of VLDL, and excessive NEFA mobilization due to aggravated NEB leads to hepatic TG accumulation and fatty liver [8,9,10,11]. Elevated blood NEFA levels accurately represent such biological conditions and thus are used as an indicator of fatty liver as well as a predictor of other related diseases [12,13,14,15,16].

Lecithin:cholesterol acyltransferase (LCAT) is synthesized in the liver and activated by apolipoprotein A-I (ApoA-I) to convert free cholesterol (FC) to cholesteryl esters (CE) in the blood [10,17,18]. Because LCAT activity decreases in cows with fatty liver, it is considered to directly reflect this pathological condition [19,20]. Cows with RP have reduced levels of ApoA-I [21,22], suggesting that RP is related to fatty liver [20,23,24,25]. Moreover, LCAT activity is a potential indicator of periparturient disease because this decrease usually precedes ketosis and milk fever [26]. In the present study, we hypothesized that LCAT activity would be a possible predictor of RP as well as NEFA concentration. To test this, we measured and analyzed serum LCAT activity in late dry cows and compared its usefulness as a predictive marker of RP with the NEFA concentration.

## 2. Materials and Methods

### 2.1. Study Design

This study was performed at a single farm with approximately 200 Holstein parous cows in Ebetsu, Hokkaido, Japan. The dairy herd consisted of 6 groups: postfresh, high-yield, low-yield, first-lactating, far-off, and close-up dairy cattle. The far-off and close-up groups were housed in free barns and the others in 2-row free-stall pens.

Dry cows were fed a total mixed ration (TMR) containing hay, grass silage, dent corn silage, beet pulp, and soybean meal with added minerals, and lactating cows were fed a TMR containing hay, grass silage, dent corn silage, beet pulp, soybeans, and cottonseed with added minerals. For both groups, the hay was timothy grass and the grass silage was a mix of timothy grass, orchardgrass, and alfalfa. Cornell-Penn-Miner Dairy software ver. 3.0 (University of Pennsylvania, Philadelphia, PA, USA) was used for designing the required ration, and the net energies per kg of dry matter were 1.55 ± 0.02 Mcal for dry cows and 1.71 ± 0.35 Mcal for lactating cows.

Of the 1187 animals that calved between February 2010 and February 2016, 915 (603 cows and 312 heifers) that underwent a routine regular health examination during the close-up stage, specifically between 2 and 21 days before their expected calving dates, were enrolled as investigated animals. Because serum NEFA is physiologically elevated from 48 h before calving [4], 80 cows that calved within 48 h of sampling were excluded. The final number of animals was 835 (549 cows and 286 primigravid heifers) with sampling dates ranging from 3 to 29 days before the actual calving dates. In the influences of calving number in seasons, the parity of dam, the calving number of male and female, there were no significant differences between the RP group and controls (*p*-value: 0.820, 0.875, 0.105, 0.118, respectively). For delivery assistance, only spontaneous deliveries and minor assistance that could be handled by one farmer were considered in this study, excluding deliveries that required two or more persons, such cases of dystocia.

This experiment was designed as a case–control study. Medical and birth records from the farm revealed that 29 of the 549 cows failed to expel the placenta within 24 h of calving and developed RP [1]. Because two animals were pregnant with twins, and thus were at increased risk of RP, they were excluded from the study [2]. In total, 27 cows remained as RP cases. The sample size was calculated using commercial software (pROC, BESE Inc., Tokyo, Japan) [27]. The numbers of samples were determined based on a case–control ratio of 1:2, with an alpha of 0.05 and 90% power. Since the number of cases was 31 and there were 62 control cows, based on the result of the calculation, 27 cows were selected as the RP group and 60 cows were extracted as controls from the enrolled cows. The 60 control cows that were clinically healthy and did not develop RP (no signs of disease until 21 days after calving) were matched with the RP group for days in milk from expected calving (mean ± SE, 95%CI, RP: −10.3 ± 0.76 d, −8.8 to −11.8 d; controls: −9.1 ± 0.51 d, −8.0 to −10.1 d, *p* = 0.177) and parity (PR: 2.93 ± 0.16, 2.60 to 3.25; controls: 2.87 ± 0.11, 2.65 to 3.08, *p* = 0.763).

Our laboratory only performed regular health examinations. Disease diagnosis and treatment were performed by veterinarians of Livestock Clinic, Hokkido Agricaltural Mutual Aid Association, at the request of the farmers. In addition, the status of calving assistance was checked by a farm manager. The respective data were shared among the three parties for farm health management.

### 2.2. Physical Monitoring and Blood Sampling

The body condition score (BCS) and rumen fill score (RFS) were evaluated as indices for physical monitoring. BCS was used as a measure of body fat storage, as described by Ferguson et al. [28]. RFS was used as an indicator of dry matter intake (DMI) and the passage rate of ingested feed [29]. Briefly, the left paralumbar fossa of each dairy animal was observed from behind to assess the degree of filling by DMI, which was scored on a scale from 1 to 5 (1 = very poor, 2 = poor, 3 = good, 4 = very good, 5 = excellent).

Blood samples were obtained from the tail vein of close-up cattle during their regular health examination between 09:30 and 10:00 just before morning feeding. Collected blood was immediately stored at 4 °C. EDTA tubes were used for NEFA concentration assay [30], sodium fluoride collection tubes for blood glucose concentration assay, and plain collection tubes for other measurements. Blood collected in EDTA and plain tubes was centrifuged at 2000× *g* for 15 min within 2 h of collection and plasma and sera after collection were stored at −20 °C until analysis.

All animals were treated appropriately following the Laboratory Animal Control Guidelines of Rakuno Gakuen University, which essentially conform to the Guide for the Care and Use of Laboratory Animals of the National Institutes of Health in the United States [31].

### 2.3. Blood Analyses

All samples were measured by colorimetric assay. Briefly, plasma NEFA, blood glucose, serum CE and FC concentrations were measured as duplicate using an automatic analyzer (BioMajesty JCA-BM2250, JEOL, Tokyo, Japan) at Kishimoto Clinical Laboratory. Serum LCAT activity was determined using a commercial kit (Anasolv LCAT, Sekisui Medical, Tokyo, Japan) as described previously [19,20,26,32].

### 2.4. Statistical Analysis

Data were analyzed using SPSS ver. 27.0 (IBM Japan, Tokyo, Japan). Normality of the blood components was established using the Shapiro–Wilk test. NEFA concentrations were log-transformed and LCAT activity values were square root transformed. Other analyzed variables were untransformed.

The analysis of blood metabolites, BCS and RFS in RP cows and controls was evaluated using a linear mixed model. The statistical model was as follows:*Y_pq_* = *μ* + *Type_p_* + *Cow_q_* + *e_pq_*
where *Y_pq_* is the observed value (NEFA, LCAT, CE, FC, glucose, BCS, RFS); *μ* is the overall mean; *Type_p_* is the fixed effect of the *p*th class of type (*p* = RP cow, control); *Cow_q_* is the random effect of the *j*th cows (*q* = 1–87); and *e_pq_* is the residual error. Data are expressed as estimated mean ± standard error. Bonferroni’s multiple comparison test was used when comparing groups.

The sample size of the case study was calculated using the commercial software mentioned above (pROC, BESE Inc., Tokyo, Japan). The chi-square test was used to compare the occurrence of RP or perinatal disease in groups. The odds ratio, sensitivity, specificity and likelihood ratio were calculated to evaluate the associations between RP and the examined parameters. The diagnostic performance of NEFA, LCAT, BCS and RF was evaluated using receiver operating characteristic (ROC) curves and measuring the area under the curve (AUC). Furthermore, binomial logistic regression was performed to analyze the association between the occurrence or not of RP and the related parameters. The occurrence or not of RP was set as the dependent variable, and NEFA; glucose, CE and FC concentrations; LCAT activity; parity; BCS; and RFS were selected as explanatory variables. Statistically significant differences were evaluated at *p* < 0.05 and a trend was assessed at *p* < 0.10.

## 3. Results

### 3.1. Occurrence of Retained Placenta (RP) and Periparturient Diseases

Of the 1187 cows (779 multiparous, 408 primiparous) that calved during the study period, RP occurred in 91 cows (7.7%). There was a trend toward a higher incidence in multiparous cows (8.6%, 67 cows, *p* = 0.095) than in primiparous cows (5.9%, 24 cows). In the multiparous cows, 43.3% (29 cows) of those with RP developed periparturient disease (ketosis, hypocalcemia, abomasum displacement, puerperal fever, metritis, and mastitis or any complication), which was a significantly (*p* < 0.01) higher rate than the 20.6% (147 cows) of the 712 non-RP cows. Of the 27 cows selected for the RP group in this study, 11 (40.7%) developed periparturient disease within 21 days postpartum.

### 3.2. Blood Metabolites and Physical Scores of Close-Up Cows

The concentrations of blood metabolites and physical scores in controls and RP cows are shown in Table 1. The concentration of plasma NEFA for the RP group was significantly (*p* < 0.01) higher than that for the control group. Serum LCAT activity of the RP group was significantly (*p* < 0.01) lower than that of the control group. There was no significant difference in the blood glucose levels or CE and FC concentrations between the RP and control groups. There was no difference in BCS between the RP and control groups. However, RFS was significantly (*p* < 0.01) lower in the RP group than in the control group.

### 3.3. Association Between Retained Placenta (RP) and Blood Metabolites

Table 2 shows the associations between the plasma NEFA concentration and serum LCAT activity with RP occurrence. The risk of RP was 2.9 times higher in animals with NEFA concentrations of ≥0.2 mEq/L than in those that had lower levels. Cows with LCAT activity of ≤500 U had a 2.9 times higher risk of developing RP than the animals with higher values. Figure 1 shows the ROC curves evaluating the relationships between the occurrence of RP and the NEFA concentration and LCAT activity. The AUCs of the NEFA concentration and LCAT activity were 0.698 (95% confidence interval: 0.579–0.817, *p* < 0.01) and 0.680 (0.555–0.805, *p* < 0.01), respectively, indicating that the diagnostic accuracy of the two serum measurements was similar.

### 3.4. Logistic Regression Analysis

Following logistic regression analysis with the occurrence of RP as the dependent variable and the NEFA concentration, parity, glucose level, CE and FC concentrations, BCS, and RFS as the explanatory variables, the NEFA concentration and RFS were selected for the final model used for prediction (Table 3). The equation obtained was 2.6801 + 0.978 × NEFA concentration (log-transformed value)−1.366 × RFS, with 74.7% predictive accuracy.

Serum LCAT activity and RFS were selected for the final model used for prediction when logistic regression analysis was performed with LCAT activity, parity, the glucose level, CE and FC concentrations, BCS, and RFS as the explanatory variables (Table 4). The equation obtained was 5.374−0.540 × LCAT activity (square root transformed values)−1.653 × RFS, with 73.6% predictive accuracy.

We further evaluated the diagnostic performance of these prediction models with ROC curves (Figure 2). In the equation where serum LCAT activity and RFS were selected, the AUC was 0.784 (0.682–0.886, *p* < 0.01). For the equation with the NEFA concentration and RFS, the AUC was 0.752 (0.646–0.858, *p* < 0.01).

## 4. Discussion

The incidence of cows with RP in this study was within the range previously reported (9–15%) [3]. However, there was a trend for the incidence of multiparous cows to be higher than that of primiparous cows, suggesting the number of parity as a risk factor. The incidence of periparturient disease on the farm in this study was 43.3% for cows that developed RP. This was higher than the 20.6% observed for non-RP cows, which is in line with a report suggesting RP as a risky pathogenesis for periparturient diseases [1]. From this actual occurrence, it can be inferred that RP is a disease with high economic losses [3].

The NEFA concentration in the late dry stage was significantly (*p* < 0.01) higher in the RP group than in the controls (Table 1), which is consistent with previous reports [16,21]. Since a high NEFA concentration is suggestive of NEB and is associated with fatty liver [8,9,10,11], it is used as a predictor for periparturient diseases [3,12,13,14,15,16].

Serum LCAT activity was significantly (*p* < 0.01) lower in the RP group than in the control group (Table 1). Similarly to the NEFA concentration, this decrease was also indicative of fatty liver because reduced LCAT activity has been demonstrated in ethionine-induced fatty liver [33], as well as in naturally occurring hepatic lipidosis [19]. Primarily, LCAT is responsible for the esterification of FC in the blood [17,23]. However, there was no significant difference in the CE concentration between the RP and control groups in our study. Most studies that compare LCAT activity and the CE concentration are limited to the postpartum period [19,32]. These studies show reduced values for both parameters, but there is no mention of an antepartum relation between LCAT activity and the CE concentration. Qu et al. did not measure LCAT activity, but they reported no difference in antepartum cholesterol between cows that developed RP and healthy ones [16], which is consistent with our findings. In contrast, Nakagawa et al. reported that cows that developed ketosis between 2 days antepartum and 9 days postpartum had reduced LCAT activity and CE concentrations before this event [26]. However, they could not find any clear relation between these changes and occurrence of the disease, as similar changes were observed in clinically healthy cows as well. Thus, although there is room for further discussion regarding predisease changes in cholesterol concentrations, the decrease in LCAT activity in the close-up stage in RP cows was confirmed.

Cows with NEFA concentrations ≥0.4 mEq/L were 5.4 times more likely to develop RP than those with concentrations of <0.4 mEq/L, and cows with LCAT activity ≤450 U were 3.6 times more likely to develop RP than those whose activity was >450 U.

The AUC of the ROC curve for serum LCAT activity was similar to that of the blood NEFA concentration (Figure 1), which suggested that LCAT activity was as useful as the NEFA concentration in predicting RP. Significant reductions in RFS were observed in RP cows (Table 1), indicating that lower DMI was a possible risk factor for the occurrence of RP.

The diagnostic performance of serum LCAT activity and other measurements in the prediction of RP occurrence was assessed by logistic regression analysis. Serum LCAT activity and RFS were selected for the final prediction model, with adjusted odds ratios of 0.583 and 0.191, respectively. This means that the likelihood of RP occurrence was inversely correlated to LCAT and RFS values. In other words, reductions in serum LCAT activity and RFS levels by 50 U and score 1, respectively, were associated with increased risks of RP occurrence of 1.7- and 5.2-fold, respectively (Table 4). Because the AUC of the ROC curve from this prediction equation (Figure 2) was bigger than that for serum LCAT activity alone (Figure 1), the contribution of not only LCAT but also RFS in predicting RP was shown to be significant. On the other hand, logistic regression analysis of the blood NEFA concentration and RFS revealed that a 0.1 mEq/L increase in NEFA and a reduction in RFS by score 1 increased the risk of RP by 2.7 and 3.9 times, respectively (Table 3). The AUC obtained using this prediction was similar to that for LCAT activity and RFS. In our recent study of predictive indicators for postpartum culling [34], the diagnostic accuracy of RFS was shown to be as good as that of serum metabolites, so further scrutiny including DMI is needed.

In conclusion, our study shows that serum LCAT activity is a valid predictor of RP in the close-up stage, similar to NEFA. Although the blood NEFA concentration accurately represents NEB, a certain level of caution is required in sampling for NEFA because it tends to fluctuate with food intake and exercise [35,36]. Thus, LCAT activity may be a more practical predictor because it is not similarly affected. In addition, careful examination of the role of RFS in the prediction of RP may be needed.

## Figures and Tables

**Figure 1 animals-14-03640-f001:**
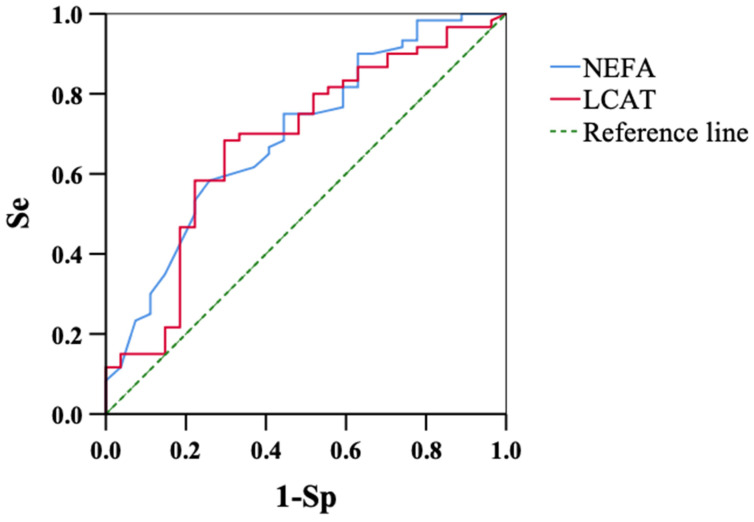
Receiver operator characteristic (ROC) curves of NEFA and LCAT for predicting the risk of retained placenta (RP). The blue and red lines show NEFA (non-esterified fatty acids) and LCAT (lecithin:cholesterol acyltransferase), respectively. The AUC of NEFA (0.698, *p* < 0.01, 95% confidence interval: 0.579–0.817) and that of LCAT (0.680, *p* < 0.01, 0.555–0.805) are similar.

**Figure 2 animals-14-03640-f002:**
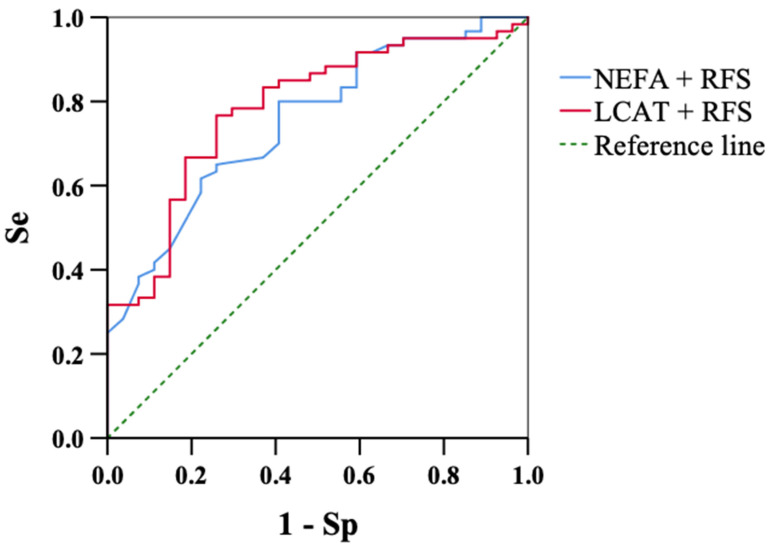
Receiver operator characteristic (ROC) curves of NEFA (non-esterified fatty acids) and LCAT (lecithin:cholesterol acyltransferase), and RFS (rumen fill score) for predicting the risk of retained placenta (RP). The blue and red lines show NEFA + RFS and LCAT + RFS, respectively. The AUC of NEFA+RFS (0.752, *p* < 0.01, 95% confidence interval: 0.646–0.858) and that of LCAT+RFS (0.784, *p* < 0.01, 0.682–0.886) are improved by adding RFS compared with each metabolite only (see Figure 1).

**Table 1 animals-14-03640-t001:** Comparison of blood metabolites and physical scores in cows with retained placenta examined in the close-up stage and controls.

Variable ^1^	Controls	Retained Placenta	*p*-Value
LCAT (U) ^2^	364.0 ± 1.4 (280.4–458.6)	167.1 ± 2.6 (93.0–262.8)	*p* < 0.01
NEFA (mEq/L) ^3^	0.154 ± 0.01 (0.134–0.178)	0.244 ± 0.01 (0.186–0.319)	*p* < 0.01
CE (mg/dL)	78.8 ± 2.1 (74.6–83.0)	74.3 ± 3.4 (67.2–81.3)	*p* = 0.29
FC (mg/dL)	19.4 ± 0.5 (18.3–20.4)	18.2 ± 0.9 (16.5–20.0)	*p* = 0.28
Glucose (mg/dL)	63.2 ± 0.4 (62.2–64.0)	61.8 ± 0.9 (60.1–63.6)	*p* = 0.13
BCS	3.32 ± 0.02 (3.27–3.37)	3.33 ± 0.05 (3.23–3.42)	*p* = 0.83
RFS	3.17 ± 0.06 (3.04–3.29)	2.81 ± 0.08 (2.65–2.97)	*p* < 0.01

^1^ LCAT: lecithin:cholesterol acyltransferase, NEFA: non-esterified fatty acids, CE: cholesteryl esters, FC: free cholesterol, BCS: body condition score, RFS: rumen full score. ^2^ Square root and ^3^ log_10_ transformations were applied to LCAT and NEFA for analysis, respectively. All obtained data were back transformed as final results. Data are expressed as estimated mean ± standard error. Numbers in parentheses show 95% CI.

**Table 2 animals-14-03640-t002:** Relevance of prepartum serum lecithin:cholesterol acyltransferase (LCAT) activity and plasma non-esterified fatty acids (NEFA) concentration in close-up stage dairy cows to the risk of subsequent retained placenta (RP).

Cutpoint	At/Above Cutpoint (%)	Risk of RP at/Above Cutpoint (%)	Risk of RP Below Cutpoint (%)	Odds Ratio	95% CI ^1^	*p*-Value	Se (%) ^2^	Sp (%) ^3^	LR ^4^
LCAT (U) ^5^									
≥550	71.3	35.5	20.0	2.2	0.7–6.7	0.158	81.5	33.3	1.2
≥500	66.7	37.9	17.2	2.9	1.0–8.8	0.049	81.5	40.0	1.4
≥450	63.2	40.0	15.6	3.6	1.2–10.8	0.018	81.5	45.0	1.5
≥400	55.2	43.8	15.4	4.3	1.5–12.1	0.004	77.8	55.0	1.7
≥350	54.0	44.7	15.0	4.6	1.6–13.0	0.003	77.8	56.7	1.8
≥300	48.3	45.2	17.8	3.8	1.4–10.1	0.006	70.4	61.7	1.8
≥250	44.8	48.7	16.7	4.8	1.8–12.7	0.001	70.4	66.7	2.1
≥200	42.5	48.6	18.0	4.3	1.6–11.4	0.002	66.7	68.3	2.1
≥150	34.5	46.7	22.8	3.0	1.2–7.6	0.022	51.9	73.3	1.9
NEFA (mEq/L) ^6^									
≥0.1	81.6	35.2	12.5	3.8	0.8–18.1	0.065	92.6	23.3	1.2
≥0.2	41.4	44.4	21.6	2.9	1.1–7.4	0.023	59.3	66.7	1.8
≥0.3	19.5	58.8	24.3	4.5	1.5–13.5	0.006	37.0	88.3	3.2
≥0.4	10.3	66.7	26.9	5.4	1.2–23.7	0.023	22.2	95.0	4.4
≥0.5	5.7	80.0	28.0	10.3	1.1–96.7	0.031	14.8	98.3	8.9

^1^ 95% confidence interval, ^2^ Sensitivity, ^3^ Specificity, ^4^ Likelihood ratio, ^5^ lecithin:cholesterol acyltransferase, ^6^ NEFA: non-esterified fatty acids.

**Table 3 animals-14-03640-t003:** Binominal logistic regression model for the association of the occurrence of retained placenta (RP) after calving with prepartum plasma non-esterified fatty acids (NEFA) concentration and other monitoring indicators.

Variable ^1^	Level	Estimate	Robust Standard Error	Odds Ratio	95% CI ^2^	*p*-Value
Intercept	-	2.68	2.079	-	-	-
NEFA ^3^	0.1 mEq/L increase	0.978	0.453	2.7	1.1–6.5	0.031
RFS	Score 1 increase	2.680	0.661	0.26	0.07–0.93	0.039

^1^ NEFA: non-esterified fatty acids, RFS: rumen fill score, ^2^ 95% confidence interval. ^3^ log_10_ transformed.

**Table 4 animals-14-03640-t004:** Binominal logistic regression model for the association of the occurrence of retained placenta (RP) after calving with prepartum serum lecithin:cholesterol acyltransferase (LCAT) activity and other monitoring indicators.

Variable ^1^	Level	Estimate	Robust Standard Error	Odds Ratio	95% CI ^2^	*p*-Value
Intercept	-	5.374	2.034	-	-	-
LCAT ^3^	50 U increase	−0.540	0.223	0.58	0.38–0.90	0.015
RFS	Score 1 increase	−1.653	0.663	0.19	0.05–0.70	0.013

^1^ LCAT: lecithin:cholesterol acyltransferase, RFS: rumen fill score, ^2^ 95% confidence interval. ^3^ Square root transformed.

## Data Availability

The data presented in this study are available upon request from the corresponding author.

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
