# Peer review of "A Case–Control Study on the Usefulness of Serum Lecithin: Cholesterol Acyltransferase Activity as a Predictor of Retained Placenta in Close-Up Dairy Cows"

_animals, 2024, doi:10.3390/ani14243640_

Round 1

Reviewer 1 Report

Comments and Suggestions for Authors

In the presented study, the activity of lecithin:cholesterol acyltransferase (LCAT) was investigated for the early detection of retained placenta (RP). However, I have many reservations about the article. The authors should emphasize more why LCAT, which forms the basis of the study, was chosen. The Introduction section of the presented article mostly focuses on negative energy balance. At this point, LCAT should be highlighted. The hypothesis of the study has not been clearly stated.

In addition, there is a significant deficiency in terms of material and method in the presented study. The collection of blood samples on a wide scale (between 2-21 days prepartum) negatively affected the reliability of the obtained data. Moreover, there are many predisposing reasons for RP. The authors did not present any data in this area. (Seasonal effect/offspring gender/differences in sperm used/offspring weights etc.)

The authors should indicate from which veins the blood samples were taken (Jugular vein? Tail vein?)

Author Response

 Thank you so much for your pertinent advice!

Reviewer 2 Report

Comments and Suggestions for Authors

The article is generally well written and presented in a manner that make it easy to follow the design and outcomes of the study. There are some points that I would like clarified, as follows:

Line 96: who conducted the assessment of the cows pre calving? Also data are presented at the beginning of the results section that alludes to periparturient disease. Who determined the occurrence of these diseases? Were cows submitted for veterinary examination if sick? Where there organised post-calving health checks?  

Line 100 : are these primigravid heifers?

Line 112 to 114:  please improve the presentation of the data here. Include the word ‘to’ rather than putting in a ‘–‘

Line 120: The reference cited here [29] is incorrect. It appears in the list as Hulsen.

Line 121: Reconsider the wording. Is ‘tension’ the correct word to use?

Line 123: Which blood vessel were blood samples collected from? – coccygeal / jugular?

Line 132: was the study submitted for ethical review?

Line 136: were samples analysed as single or duplicate samples?

Line 136 to 139: What is the detection method used?

Line 144: what was the reason for the transformation?

Line 142: was day included in your analysis?

Line 155: Define the groups.

Line 177: Do you mean ‘udder infection’ (mastitis) or ‘udder inflammation’? If the latter, how was this determined?

Line 184: what are the data shown? Mean +/- sd or sem?

Line 187: please define all of the abbreviations used in Table 1, in the footnotes e.g. LCAT, CE, FC etc..

Line 191: the RP risk is also significant where NEFA concentrations are ≥0.2 mEg/L, from Table 2?

Line 201: please define all of the abbreviations used in Table 2, in the footnotes to make the tables and figures standalone. Also Figure 1, Table 3, Table 4 and Figure 2.

Line 246: Is this what you expected?

Line 250: wording of the discussion here – is this indicating economic loss? – or that there is a problem on the farm

Line 252: Why are cows in NEB more likely to develop RP?

Line 272: Why are cows with high NEFA likely to develop RP?

Line 304: author contributions need to be confirmed

Author Response

 Thank you so much for your suggestive advice!

Reviewer 3 Report

Comments and Suggestions for Authors

The prediction of periparturient diseases is a current hot topic in research. This is a meaningful study that elucidates the biological significance of LCAT in comparison to NEFA and .I think that this work should be publication. Satoh and colleagues investigated the predictive ability of serum lecithin: cholesterol acyltransferase (LCAT) for retained placenta in close-up dairy cows using a case-control study design. LCAT, a key regulatory enzyme in cholesterol metabolism, is thought to be associated with the occurrence of diseases in dairy cows. Through logistic regression modeling and ROC analysis, the study confirmed that LCAT possesses strong predictive capabilities. Predictive biomarkers for diseases in transition dairy cows have become a current research focus, and the study by Satoh et al. offers new insights into the prediction of retained placenta. When combined with previously studied non-esterified fatty acids (NEFA), LCAT demonstrates a higher predictive ability alongside rumen fill score. However, this investigation was conducted with a small sample size, and validation work was not included, a limitation acknowledged by the authors in the manuscript. The references cited in the article are appropriately selected and include recent research papers. Regarding tables, the authors should implement three-line tables instead of single-line tables, which requires modification. In addition, there is a vague area in the article: the representation of the data should be clarified. Is it mean±SD or mean±SE?

Author Response

(The authors gave the same response as above.)

Round 2

Reviewer 1 Report

Comments and Suggestions for Authors

No suggestions

Author Response

Thank you so much for your pertinent comments!